



# Innovative L-band electron paramagnetic resonance investigation of solid-state pouch cell batteries

Charles-E Dutoit[1,2], Raffaella Soleti[3,4], Jean-Marie Doux[5], Vincent Pelé[5], Véronique Boireau[5], Christian Jordy[5], Simon Pondaven[2,6], and Hervé Vezin[1,2]

[1] Université Lille Nord de France, CNRS, UMR8516, LASIRE, 59655 Villeneuve d'Ascq, France
[2] Centre de Résonance Magnétique Electronique pour les Matériaux et l'Energie, 59655 Villeneuve d'Ascq, France
[3] Université d'Angers, Inserm UMR1307, CNRS UMR6075, Nantes Université, CRCI2NA, F-49000 Angers, France
[4] Université d'Angers, SFR ICAT, Plateforme RPE, F-49000 Angers, France
[5] SAFT, Direction de la Recherche, 111 Boulevard Alfred Daney, 33074 Bordeaux, France
[6] TotalEnergies OneTech R&D, Centre de Recherche de Solaize (CRES), chemin du canal, BP 22, 69360 Solaize, France

**Correspondence:** Charles-E Dutoit (charles.dutoit@univ-lille.fr)

**Abstract.** Usually, the conventional electron paramagnetic resonance (EPR) spectroscopy and imaging use a microwave cavity operating at X-band, i.e. with an excitation frequency of around 9.6 GHz, and remains the most popular mode in the magnetic characterization of lithium batteries to date. We provide here the first low-frequency EPR investigations for monitoring the metallic lithium structures in the solid-state pouch cell batteries. We show that L-band, i.e. a microwave frequency of around 1.01 GHz, is an invaluable method to probe in depth the electrode components through a standard pouch cell using aluminum laminated film for packaging without opening the battery. These results offer a new approach for monitoring the nucleation of micrometric and sub-micrometric lithium particles such as dendritic lithium structures which is an important step in the development of reliable solid-state batteries.

## 1 Introduction

Lithium-ion batteries offer high energy densities making them appealing for a wide range of applications. Conventional Li-ion cells are composed of a Li-containing oxide as a positive electrode material and graphite as a negative electrode material. The Li$^+$ ionic transfer between the electrodes is done through a liquid electrolyte flooding both electrodes and a separator keeping both electrodes apart (Armand and Tarascon, 2008). One of the attractive cathodes for commercial use is made up of a lithium containing metal oxide such as lithium, nickel, manganese, cobalt, oxide commonly known as NMC (Santhanam and Rambabu, 2009; Yaqub et al., 2014; Leng et al., 2017). These elements form a stable structure to hold the lithium-ions when the battery is in a discharged state. With the goal of improving the energy density and safety of cells, the replacement of the liquid electrolyte by a solid compound (gel, polymer, ceramic or their combination) is being studied in solid-state batteries (Agrawal and Pandey, 2008; Hatzell et al., 2020). While most commercial Li-ion cells are hosted in rigid structures (cylindrical and prismatic cells), softer casing made of laminated aluminum and polymer layer are also being developed for the pouch cell packaging. This format can easily be made into different shapes and sizes while benefiting from a lightweight packaging which is a serious advantage for mobility applications. Considering the versatility of the pouch cell format, it is also adapted for





the solid-state battery. Furthermore, the packaging is not a rigid enclosure and a special attention is needed because of safety concerns during short-circuits or overcharge events as the pressure buildup can cause cell swelling (Chen et al., 2021). Such short-circuits, usually caused by the nucleation of metallic lithium aggregates during lithiation processes can pose a serious
risk of explosion. Early non-destructive detection of lithium growth is mandatory to avoid such risk in this cell format. In this work, using continuous-wave L-band electron paramagnetic resonance (cw-EPR) spectroscopy and imaging, we report, for the first time, the possibility of analyzing the metallic lithium structures in the solid-state pouch cell batteries directly through the standard aluminum laminated film. One of limitations in the EPR characterization of an electrochemical cell is the cell design which have to be compatible with the standard resonator dimension. In previous EPR measurements, we designed a millimetric
cylindrical electrochemical cell specially adapted for X-band spectrometers (Salager et al., 2014; Sathiya et al., 2015; Dutoit et al., 2021). It should be noted that all previous in situ EPR results bave been performed on electrochemical cell models using a Kapton film, PET/EVA film or other types of EPR silent polymers which are not a standard packaging for batteries (Geng et al., 2021). Herein, we demonstrate that the electrode materials enveloped in an Al-based pouch film used as a standard packaging materials for batteries (Park et al., 2021) can be characterized without opening the battery. This result offers a new approach
paving the way for operando characterization of commercial pouch cell batteries.

## 2    Experimental details

### 2.1    Samples

Two types of cells were assembled using the same positive electrode and separator layers with either a Li-metal or Si-based negative electrode. An argyrodite-type sulfide electrolyte was selected as the solid electrolyte (SE). The positive electrode,
separator and Si-based negative electrode were all prepared using a wet process using isobutyl isobutylene and xylene solvents with PVdF-based copolymer as binders for the electrodes and an in-house specific binder for the separator. All the materials were handled and processed in Ar-filled glove box. For the positive electrode, NMC and SE powders were dispersed in a binder gel solution using a planetary mixer. A slurry containing NMC, SE and PVdF was casted on a carbon-coated Al foil with a doctor blade with a 20.5 mg/cm$^2$ loading. For the Si-based negative electrode, Si ($\mu$m size), SE and Super P carbon
were dispersed in the binder gel solution. A slurrry containing Si, SE, C and PVdF was casted on a carbon-coated Cu foil with a doctor blade and a 2.9 mg/cm$^2$ loading was obtained. For the separator, the SE powder was dispersed in the binder gel solution; and the obtained SE:binder (weight ratio 97:3) slurry was casted on a release-type PET film with a doctor blade. All the layers were dried at room temperature in the glove box. The uncalendared separator had a thickness of 130 $\mu$m ($\pm$10 $\mu$m) and could be peeled off from the PET film to recover a self-standing solid electrolyte layer. The Li metal had a thickness of
60 $\mu$m. The different layers were punched. The positive and negative electrodes were welded to tabs and set in the PP-coated aluminum laminated, before sealing the obtained pouch. The pouch was compacted under 300 MPa using a cold isostatic press to densify the layers and improve the electrode/separator contacts. The voltage was then controlled to ensure the pouchs were not short-circuited.



## 2.2 Electron paramagnetic resonance spectroscopy

Continuous wave (cw) EPR experiments were performed using an L-band Bruker spectrometer operating at microwave frequency of around 1.01Ghz. EPR measurements were made at room temperature in a cylindrical microwave loop-gap cavity. The microwave power applied into the loop-gap cavity was set to 36 mW. The modulation amplitude of the magnetic field was taken at 0.3 mT. Other spectrometer settings were: sweep width, 200 G; conversion time, 40 ms; number of scans, 1; sweep time, 40.96 ms. Simulations were done using the EasySpin package for MATLAB (Stoll and Schweiger, 2006).

## 2.3 Electron paramagnetic resonance imaging

EPRI measurements were performed with an L-band Bruker imaging spectrometer equipped with a three-axes gradient coil set with gradients along the x-axis (perpendicular to **Y** and $\mathbf{H}_0$), y-axis (perpendicular to **X** and $\mathbf{H}_0$) and z-axis (along $\mathbf{H}_0$). Images were done with a gradient strength of 42 G/cm and a field-of-view of 30 mm. The high resolution images were reconstructed with a size of 512x512 pixels which gives a pixel size of 150 $\mu$m. The recorded projections under gradient were deconvoluted

from a signal obtained without gradient. Finally, the spatial-spatial images were obtained after a filtered back-projection.

## 3 Results and Discussion

Low frequency EPR experiments were carried out to probe electrode materials of two solid-state pouch cell batteries. In situ pouch cells were measured at the pristine state without opening the batteries. Figure 1 shows the real dimensions of the NMC‖$Li^0$ pouch cell battery and its orientation inside the loop-gap cavity. Usually, **Z** axis corresponds to the static magnetic

field direction (**B**). The pouch cells, 65x38 mm, are placed in the microwave loop-gap cavity (diameter, ∼40 mm; length, ∼70 mm) such as the battery plane is oriented along the static magnetic field, i.e. perpendicular to the **Y** axis. However, probing conduction electrons in metallic structures is not trivial due to a limited penetration of the microwave in the conductor also knwon as the skin depth. Indeed, it is well known that EPR line shape of metallic conductors is sensitive to the metal-thickness and the skin depth. In an EPR experiment, the electromagnetic field penetrates only the skin depth $\delta_{mw}$ and only

the spins located in this area are irradiated. Furthermore, this phenomenon depends on the spin coherent time $T_2$, in which electron travels on a specific distance $\delta_e$, named spin depth. In metallic lithium structures, $T_2 \sim 10^{-9}$ s and then when the metal thickness is much higher than the skin depth, the EPR spectrum exhibits an asymmetric line shape characterized by the ratio A/B. This skin depth can be easily calculated from the equation defined by:

$$\delta_{mw} \propto \sqrt{\frac{\rho}{f}} \tag{1}$$

where $\rho$ is the metallic resistivity and f is the microwave frequency applied inside the cavity. It can be seen that the skin depth is proportional to $1/\sqrt{f}$. However, the physical size and the conductive behavior of the standard laminated aluminum packaging lead an additional challenge for monitoring electrode materials. Indeed, placing a large metallic conductor inside a standard X-band microwave cavity can cause serious perturbations of the dielectric making it impossible to tune the cavity.





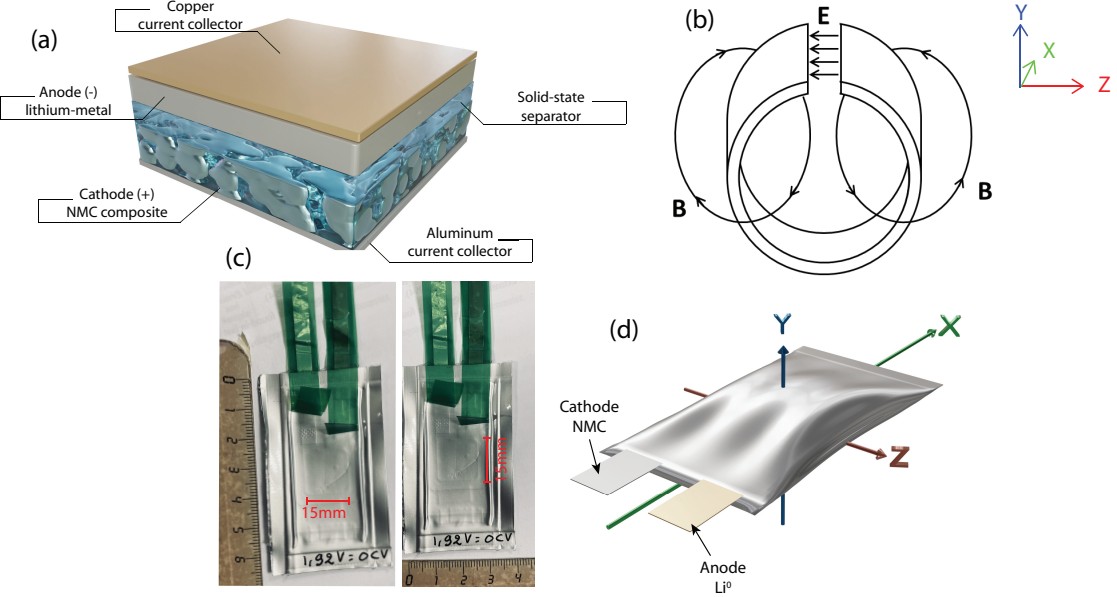

**Figure 1.** Examples of pouch cells studied in this work. (a) Schematic representation of the stacking used in our solid-state battery NMC∥Li$^0$. (b) Sketch of a microwave loop-gap cavity used for L-band EPR spectroscopy. The electrical field (**E**) is confined to the capacitive gap whereas the magnetic field (**B**) is confined inside the inductive loop. (c) Digital photos of the NMC∥Li$^0$ pouch cell. (d) Schematic representation of the pouch cell with its orientation inside the microwave loop-gap cavity.

Nevertheless, getting an optimal tuning is essential to perform EPR measurements. Consequently, we turned to L-band EPR spectroscopy and imaging which have a large loop-gap cavity adapted for standard solid-state pouch cell batteries, i.e. without specific battery preparation needed for X-band. In a such loop-gap cavity, the electrical field **E** is confined to the capacitive gap and the magnetic field **B** inside the inductive loop. As a result, the dielectric is slightly perturbed by inserting the metallic sample (see Figure 1(b)). Furthermore, at L-band, the skin depth increases compared to X-band giving a better microwave penetration through the metallic body (∼3 $\mu$m for aluminum at 1 GHz).

Figure 2(a) gives the L-band EPR spectrum of the NMC∥Li$^0$ pouch cell recorded at room temperature and in continuous wave through the standard aluminum laminated packaging. This signal exhibits a Dysonian line shape usually found for conduction electrons (Dyson, 1955; Feher and Kip, 1955). To obtain more information about the nature of this metallic complexes like the g-factor and the linewidth, we have simulated the spectrum with a phase shifted Lorentzian function, i.e. a sum of Lorentzian shaped absorption and dispersion functions, modeling the Dysonian shape (Gourier et al., 1989; Wandt et al., 2015). It can be seen that the EPR spectrum is characterized by: (i) a single line centered at a g-factor of around 2.004±2.10$^{-3}$, (ii) a peak-to-peak linewidth of 0.3 mT, (iii) an asymmetric ratio A/B around 3.7, as expected for metallic lithium structures. It is worth noting that the skin depth is around 4 $\mu$m for Li$^0$ complexes at 1 Ghz. This result indicate that the metallic structure size detected is much higher than 4 $\mu$m consistent with the metallic electrode dimensions. As the solid-state NMC∥Li$^0$ battery is composed of





a metallic lithium electrode, we may compare the spectral signature relative to a non-metallic battery taken as reference. The
result is given in Figure 2(b) where lithium metal anode is replaced by Si-based electrode. As expected, initially the solid-state
NMC‖Si-based battery, which contains $Li^+$ (EPR silent), $Ni^{2+}$ (S=1), $Ni^{3+}$ (S=1/2), $Mn^{4+}$ (S=3/2), $Co^{3+}$ (diamagnetic) and
$O^{2-}$ (EPR silent), gives no metallic lithium spectrum at the pristine state. However, at room temperature NMC exhibits an EPR
signal centered at a g-value of around 2.00 with a linewidth of around 22 mT arising from $Mn^{4+}$ ions. The presence of $Ni^{2+}$
and $Ni^{3+}$ in the NMC sample is known to create an additional spectral broadening of the $Mn^{4+}$ peak. The manifestation of
the $Mn^{4+}$ is represented by a distortion of the baseline clearly visible for NMC‖Si-based cell but invisible for NMC‖$Li^0$ cell
due to the intense $Li^0$ signal. Nevertheless, a single EPR peak is observed, displaying a g-factor near the free electron g-value
($g_e$=2.0023), originates from $SiO_2$ defects of the anode.

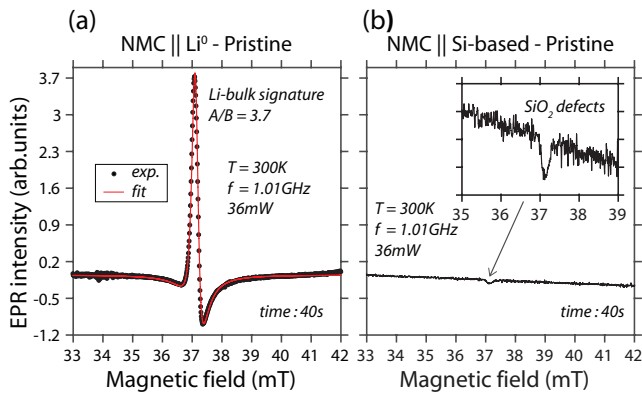

**Figure 2.** Continuous wave L-band EPR spectra of two solid-state batteries recorded through the aluminum laminated pouch cell at room
temperature and at the pristine state: (a) NMC‖$Li^0$; (b) NMC‖Si-based battery.

In order to verify that the metallic lithium spectrum does not originate from an impurity inside the pouch cell, we turned to
the spatial-spatial imaging of the NMC‖$Li^0$ solid state battery to localize the anode part. As we saw previously, metallic lithium
structures give a relatively sharp spectrum with a linewidth of around 3 G (with 0.1 mT = 1 G). Using a gradient strength of
42 G/$cm^2$ and thanks to the sharp line of $Li^0$, high resolution EPR images in the spatial planes XZ, YZ and YX respectively are
expected. After placing the pouch cell directly in the center of the microwave loop-gap cavity and gradient coils, we recorded
each image at room temperature. The time needed for each spatial-spatial image is around 60 min with a pixel size of 150 $\mu$m
and a field-of-view of 30 mm. Figure 3 presents the EPR images for a NMC‖$Li^0$ solid-state pouch cell battery. For more
clarity, we chose to reconstruct the spin concentration using a contour map with the red contour for the highest amplitude
and the blue contour for the lowest amplitude. Contrary to recent investigations using the X-band EPR imaging to monitoring
electrochemical batterie and for which the packaging was made with a non-standard material, such as a a Kapton film (Geng
et al., 2021; Kang et al., 2024), the L-band image contains the spin distribution of electrode materials directly through the
standard aluminum laminated pouch cell. The $Li^0$ anode part of the pouch cell is clearly visible and appears with a square
shape (15 mmx15 mm) in the center of image. In the limit of the resolution of an L-band EPR spectrometer, this shape and its





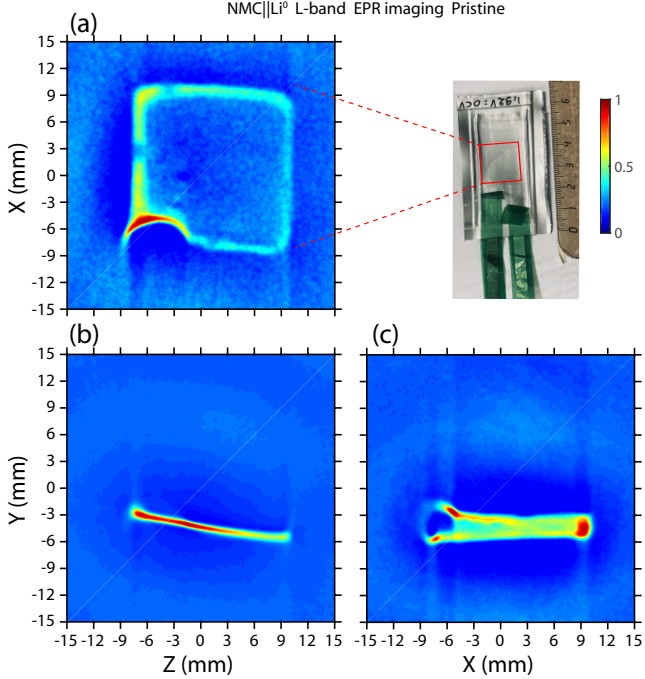

**Figure 3.** In situ L-band EPR imaging of a solid-state pouch cell NMC∥Li$^0$ battery recorded using the spatial-spatial detection scheme and in the XZ plane (a), YZ plane (b) and YX plane (c) respectively.

dimension are similar to the real size of the anode. However, a slight variation of the EPR intensities is observed between the edge and the middle of the anode. As shown by Niemöller and coworkers, the EPR signal of the thick and flat metallic electrode (much thicker than the skin depth) gives a higher apparent intensity at the edge which hides the weaker signal in the middle part of the lithium foil. The physical origin of this contrast, characterized by extreme sensitivity of the edge to the microwave field,
125 comes from a local variation of the microwave field caused by shielding and/or Eddy current effects (Niemöller et al., 2018). Nevertheless, in the case of sub-micrometric Li-metal nucleation, such as mossy/ dendritic lithium, this contrast will not be observed owing to the small size of such particles for which the electromagnetic field penetrates the whole metal. Furthermore, in Figure 3(a), an intense curved shape localized at X=-6 mm and Z=-6 mm corresponding to the highest amplitude can be observed. This result indicates a local defect much more sensitive to the microwave field which corresponds to the interface
130 between the lithium electrode and the copper current collector. Finally, images in the YZ and YX planes provide evidence that battery alignment inside the microwave loop-gap cavity is essential to reduce errors and misinterpretations. It can be seen that our solid-state pouch cell is slightly tilted in the loop-gap, which may also explain the apparent amplitude variation. Nevertheless, a three-axis field gradient associated with an L-band EPR spectrometer will enable us to locate metallic lithium structures through a standard packaging for batteries with a better spatial accuracy and then will improve the knowledge of
135 Li-metal nucleation processes.

**MAGNETIC RESONANCE**
Open Access Discussions

## 4 Conclusions

In conclusion, NMC‖Li$^0$ and NMC‖Si-based solid-state pouch cell batteries have been studied by continuous-wave (cw) L-band EPR spectroscopy and imaging. We provide first evidence of EPR characterization for pouch cell batteries using a standard aluminum laminated packaging. We observed a metallic lithium EPR spectrum and a SiO$_2$ defects signal through the aluminum laminated film. An advantage of conventional cw-EPR spectroscopy at L-band is that a centimeter size pouch cell using an aluminum laminated film material can be analyzed without preparation, without using a home-made cell specially designed for EPR and without opening the electrochemical battery.

For future studies, an operando EPR analysis of standard pouch cell batteries for solid-state or Li-ions technologies will provide additional information about the degradation processes and Red/ox evolution by probing electrode materials directly through a standard aluminum laminated film forming the pouch cell. We think our new approach would provide valuable insight for the battery community in terms of formulation, optimization and quality control of standard and commercial batteries.

*Code availability.* Matlab is a commercial software from MatlabWorks and the Easyspin package is available from https://easyspin.org/

*Data availability.* The dataset that support the findings of this investigation is available here: https://doi.org/10.5281/zenodo.14034164.

*Author contributions.* C-.E.D., H.V. and S.P. designed the project. C-.E.D. and H.V. performed EPR measurements and interpreted the results. RS provided the technical support. J-.M.D., V.P., V.B. and C.J. provided the solid-state pouch cells. The manuscript was written by C-.E.D. through contributions of all authors. All authors have given approval to the final version of the manuscript.

*Competing interests.* At least one of the (co-)authors is a guest member of the editorial board of Magnetic Resonance. The authors have no other competing interests to declare.

*Financial support.* This research has been supported by the Centre National de la Recherche Scientifique (CNRS) and by the TotalEnergies Hybrid Storage R&D program. Financial support from the joint research unit CR2ME (grant no. LS235324).

*Acknowledgements.* This work received funding from the Haut-de-France region through the STaRS program, from the Centre National de la Recherche Scientifique (CNRS) and from TotalEnergies Hybrid&Storage R&D program under the joint laboratory CR2ME (Centre



de Résonance Magnétique Electronique pour les Matériaux et l'Energie). We thank SAFT for providing the solid-state pouch cell batteries studied in this work.



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
