# Peer review of "Innovative L-band electron paramagnetic resonance investigation of solid-state pouch cell batteries"

_Magnetic Resonance, 2024_

## Author Comment (AC1)

**Point-by-point response to reviewers and editor for the manuscript:**

*Innovative L-band electron paramagnetic resonance investigation of solid-state pouch cell batteries,* by CE. Dutoit, R. Soleti, JM. Doux, V. Pelé, V. Boireau, C. Jordy, S. Pondaven and H. Vezin.

*This paper used a commercial Bruker L-band (ca. 1 GHz) EPR imager to measure a pouch cell lithium ion battery. This image will stimulate extensive applications of L-band EPR to actual batteries and their performance. The paper, however, requires revision to be more comprehensible about the materials used. As written, it would be difficult for anyone to reproduce this work without direct interaction with the lab that did the preparations.*

We sincerely thank the reviewer for his detailed reading of our manuscript, which allows us to correct an error and to address some points that need to be clarified.

**Comment #1:** *The battery described in this work uses a solid electrolyte and lithium anode. Solid electrolyte LiB are a still developing technology. The majority of LiB in use in personal devices and automotive applications have a liquid electrolyte. There are commercial sources of aluminum encased pouch cells with liquid electrolyte. Could the authors to comment on the feasibility of repeating this experiment with liquid electrolyte LiB? It would have an immediate tie in to the most widely deployed type of pouch cell LiB.*

**Response:** *Thank you for this comment which is absolutely right. Li-ion technology is mainly used in commercial batteries to date. However, our first investigation was performed on two solid-state pouch cell batteries because our initial project was focused on this technology. Following this work, we turned to L-band EPR analysis of lithium-ion pouch cell batteries sealed inside a standard Al-based packaging. It is worth noting that electrochemical performance and EPR of Li-ion cell (non-standard packaging) is well established. We investigated the (de)lithiation process of a graphite anode during cycling of a $LiFePO_4$/graphite battery as case study to demonstrate the feasibility of using operando L-band EPR. The results (confidential and which will be the subject of a future paper) shown the nucleation of an EPR peak of $Li_xC_6$ during charge which drops to 0 upon discharge. Such a finding is consistent with previous X-band investigations of the graphite (de)lithiation and opens the possibility of using operando L-band EPR as a new device for monitoring redox process.*

**Comment #2:** *Did they select a lithium anode because they thought sensitivity issues would prevent them from directly observing the cathode material EPR at 1 GHz? Typically, the anode is a graphite-based material. There are stated to be increased safety concerns for the solid lithium anode in terms of thermal runaway.*

**Response:** *First, we choose a $Li^0$ as the anode owing to its high energy density. Secondly, the main idea here was monitoring electrode materials wrapped inside an industrial aluminum-based pouch cell. Lithium metal is easily detected by EPR spectroscopy and appears to be the material of choice in preliminary studies. Furthermore, initially we expected perform operando measurements to detect the nucleation of sub-micrometric lithium particles (like dendrites). However, cycling all-state batteries is not trivial and requires some precise controls. From our good results on the detection of metallic lithium structure through Al-based pouch cell, we performed an operando investigation on a $LiFePO_4$ / graphite battery (data not shown and which will be the subject of a future paper). We observed, in real time and real conditions, the reversible graphite (de)lithiation showing the good sensitivity of our approach.*

***Comment #3:*** *The authors should add a section discussing the implications of this first experiment. How would they expect the lithium EPR signal to change as a function of ageing in this pouch cell? What do they think they would see with a graphitic anode pouch cell?*

***Response:*** *In the case of pristine battery with Li-metal as anode, the EPR spectrum expected is asymmetric and is well defined by a Dysonian function. The reason comes from the skin depth, and then from the metal thickness, which determine the EPR line shape. When the metal thickness is much higher than the skin depth the EPR spectrum appears asymmetric. It's the present case at the pristine state for a Li-metal anode. However, upon cycling, some sub-micrometric and/or nanometric lithium particles, such as dendrites, can be formed. Consequently, the EPR reveals a symmetric and intense line shape as reported in the literature with a X-band spectrometer (Niemöller et al. doi: 10.1038/s41598-018-32112-y). At L-band, similar results are expected.*

*As suggested by the reviewer, we added the following sentences (in red) on page 7.*

*"As reported in the literature, during electrochemical cycling, an intense and symmetric EPR spectrum can appears. This feature is the manifestation of an alteration of the metallic anode surface which creates sub-micrometric lithium aggregates. However, in the present work, we did not perform operando measurements and we did not observe an EPR signature of sub-micrometric lithium aggregates."*

*For the graphite anode, we observed the formation of lithiated graphite complexes during charge which disappears during discharge showing the reversible behavior of the graphite lithiation.*

***Comment #4:*** *The EPR audience may not know why these pouch cells are important compared to steel-cased cylindrical cells. For instance, the flexibility in packaging allows pouch cells to be tailored to specific space constraints better than rigid cylindrical cells. The authors don't need to "take down" cylindrical cells, but they could spend a little time answering the "so what?" question, which also helps convey the importance of this first measurement.*

***Response:*** *Thank you for this suggesting. Two sentences have been inserted (in red) on pages 1-2 of the revised paper to clarify these points:*

*"Owing to a lightweight aluminum packaging, pouch cells can be used in small portable electronics such as drones, mobile and so one. Compared to rigid cylindrical cells, they offer a good weight to energy ratio."*

***Comment #4:*** *What is the composition and availability of NMC, SE, the liquid electrolyte, the pouch batter studied, etc. "An argyrodite-type-sulfide electrolyte" is too vague to be useful.*

***Response:*** *NMC= NMC811 ($LiNi_{0.8}Mn_{0.1}Co_{0.1}O_2$) very commonly used by cathode supplier like Umicore for example. SE: $Li_6PS_5Cl$ can be supplied by NEI. Pouch made of aluminum laminate (120μm) containing one polyamide layer and one polypropylene layer (supplier: MTI for example).*

*These details are now included in the experimental section on pages 2-3 (in red).*

***Comment #5:*** *What does it mean that "the different layers were punched"?*

***Response:*** *The different layers were punched means that electrodes and electrolyte layer were cut with a special tool which cuts the layers at the final shape (see an example of a punch : http://www.hohsen.co.jp/en/products/detail.php?id=116)*

**Comment #6:** *It is not evident how "the voltage was then controlled to ensure the pouches were not short circuited."*

**Response:** *After the cell is assembled, the voltage is measured with a voltmeter and it was checked that the voltage was above 2V to ensure that there was no short circuit.*

*We added this detail on page 3, lines 56-57 (in red).*

**Comment #7:** *Presumably the resonator used was one produced by Bruker. The model/part number should be stated.*

**Response:** *Yes, the microwave resonator used in our work is a bridge loop-gap resonator produced by Bruker (L Band_BLGR_36 mm_ E1978).*

*The model/part number is now added in the experimental section (page 3, line 61 in red).*

**Comment #8:** *What do the authors mean by "can cause serious perturbations of the dielectric"? The context of the statement suggests that they really mean that the conductivity of the metal cell distorts the microwave magnetic and electric field distributions. This is repeated on page 4.*

**Response:** *Usually, a large metallic sample placed in a X-band microwave cavity, which is simply a metal box, creates a perturbation of the electromagnetic field inside the resonator owing to its high electric conductivity. Such a distortion makes its sometimes difficult to tune the cavity because of the broadening of the "dip" creates by the reflected microwave power. That is the reason why, in our previous study at X-band, we used a homemade electrochemical cell EPR silent and two metallic current collectors with a millimeter-sized which do not affect the electromagnetic field.*

*At L-band, the microwave cavity shape is different (loop-gap shape). Furthermore, we used an electrochemical pouch cell which is much larger than the homemade X-band EPR cell. As a consequence, the current collectors are also much larger. Although the metallic packaging, and the both current collectors create a weak perturbation in the loop-gap resonator, which is challenging here, we were able to tune the cavity and observe an EPR spectrum from the $Li^0$ anode.*

**Comment #9:** *The statement in line 76 that $T_2$ is approximately $10^{-9}$ s is inconsistent with the reported results. $10^{-9}$ s would correspond to about 65G peak-to-peak derivative spectrum. They report 3 G on the next page. In addition, the paper should provide references for the relaxation times.*

**Response:** *Thank you for this comment, which is right. We have measured the spin-spin relaxation time $T_2$ and the spin-lattice relaxation time $T_1$ of an Li-metal anode using FID T2 decay and FID inversion recovery at X-Band on a Li-Li symmetric battery.*
*We found $T_1=T_2$ and a value around $10^{-8}$s for the bulk lithium foil as after polarization cycles, $T_2$ and $T_1$ of microstructures are in range of $10^{-6}$ s.*
*The $T_2$ value is now corrected in the revised manuscript (page 3, line 81 in red).*

**Comment #11:** *The caption of figure 2 should include the microwave $B_1$ at the sample, and the modulation amplitude and frequency used.*

**Response:** *Yes, thank you, we added the modulation amplitude, frequency and microwave power used in the caption of Fig.2 (page 5).*

**Comment #12:** *The data described in lines 100-105 are not otherwise presented in the paper, and there is no mention of how they were measured. Possibly the information is from a report in the literature and references should be given.*

*Response:* Your remark is absolutely correct. Lines 100-105 describe the EPR result shown in Fig 2(b). In the literature we find a variety of study showing the EPR spectrum of the NMC material. Indeed, Mn$^{4+}$ gives an EPR peak with a linewidth of around 22mT and Ni$^{2+}$/Ni$^{3+}$ create a linewidth broadening. We are agreeing with the reviewer that lines 100-105 must to be clarify.

*We modified lines 100-105 (in red) by on page 5 and included a reference in the revised manuscript (ref Tang et al., 2017, page 10).*

*"However, as reported in the literature (Tang et al., 2017), at room temperature NMC material exhibits an EPR signal centered at a g-value of around 2.00 with a linewidth of around 22 mT arising from Mn$^{4+}$, Ni$^{2+}$ and Ni$^{3+}$. The manifestation of the NMC is represented by a distortion of the baseline. This distortion is clearly visible in the NMC∥Si-based cell but invisible in the NMC∥Li0 cell due to the intense Li$^0$ signal ".*

**Comment #13:** The paper should report and comment on problems encountered in this imaging measurement and solutions developed. The primary value of this paper is to say "yes, you can study these metal pouch batteries with commercial instruments." It is of value to publish as more than a claim to priority only if it contains proper information to make it reproducible.

*Response:* We think the reviewer for this comment.

*Up to now, most of the experiments have been carried out in X-Band on model batteries which, while they have the merit of providing a good understanding of materials in terms of redox, SEI formation and dendritic lithium formation, are clearly a long way from an industrially-produced battery. This work is a first step that shows the feasibility of a commercial device for which we will improve the environment so as to be able to study an entire cell in operando.*

*No problems were encountered with image acquisition. While it is possible to acquire an EPR spectrum, imaging is only a variant of EPR spectrum acquisition, since it involves encoding the signal. The strength of the 40G/cm gradients does not disturb the position of the echo during acquisition. This would perhaps be different with gradients similar to the X band, i.e. 200G/cm.*

---

## Author Comment (AC2)

**Point-by-point response to reviewer and editor for the manuscript:**

*Innovative L-band electron paramagnetic resonance investigation of solid-state pouch cell batteries,* by CE. Dutoit, R. Soleti, JM. Doux, V. Pelé, V. Boireau, C. Jordy, S. Pondaven and H. Vezin.

The authors thank the reviewer for his detailed reading of our manuscript.

**Reviewer 2**

*The authors describe EPR spectroscopy and imaging at L-band of a slid-state battery in a pouch cell format. They demonstrate that it is possible at this frequency to detect a signal even if the cell is placed inside a laminated pouch bag, corresponding to a format that is also used for standard lab experiments or some commercial batteries. While this result is valuable and publication in Magnetic Resonance would be justified, the current state of the manuscript represents various shortcomings that need to be addressed. In particular, the conclusion that "an L-band EPR spectrometer will enable us to locate metallic lithium structures through a standard packaging for batteries with a better spatial accuracy and then will improve the knowledge of Li-metal nucleation processes" is not fully substantiated, since in the presented data only signal is unambiguously discernible that originates from the boundary of the cell. It is not possible to distinguish if Li metal from the center of the cell leads to an observable signal at all.*

*Comment#1: Experimental details are currently insufficient for other laboratories to reproduce these experiments. The exact specification of the used separator and the origin of the different commercial materials should be provided.*

**Response:** *Thank you for this suggestion. NMC= NMC811 ($LiNi_{0.8}Mn_{0.1}Co_{0.1}O_2$) very commonly used by cathode supplier like Umicore for example. SE: Li6PS5Cl can be supplied by NEI. Pouch made of aluminum laminate (120μm) containing one polyamide layer and one polypropylene layer (supplier: MTI for example). These details are now included in the experimental section on pages 2-3 (in red) of the revised manuscript.*

*Comment#2: 36 mW is a high power level for such a narrow signal, which would generally lead to significant saturation effects. The authors should specify the attenuation caused by the pouch bag. Also, does this high power level cause local heating in the cell?*

**Response:** *We thank the reviewer for this important point. The EPR spectrum of the Li-metal anode is possibly saturated by using a high power, here 36mW, but such power level does not affect the origin of the EPR spectrum observed. In our work, we did not perform an operando measurement for which a preliminary investigation of saturation effects is necessary for monitoring the spectral intensity evolution. Here, we only provide the feasibility to use L-band EPR spectrometer to observe Li-metal structures through a standard and Al-based pouch cell battery. In the case of operando measurements or to quantify lithium aggregates, a small power level will be applied to the sample.*

*Please, as suggested by the reviewer, we provide below the EPR spectrum of the pristine pouch cell battery NMC811||Li-metal recorded at 6mW displaying the same general pattern (bulk Li-metal) as the one recorded at 36mW.*

[Figure]

*During our measurements, we did not observe a local heating in the cell.*

*Comment#3: A modulation amplitude of 0.3 mT is too high if the main signal has a width of 0.3 mT and, more importantly, potentially narrower signals may also occur. Please provide a spectrum recorded with a narrower modulation amplitude to exclude overmodulation effects.*

**Response:** *We are agreeing with this comment. In our work, we have deliberately chosen to over modulated the EPR signal in order to clearly observe the Li-metal signature. Initially, we had no idea about the L-band EPR line shape, neither the microwave frequency adapted to penetrate the Al-based pouch cell. Consequently, in our first investigation we recorded the EPR spectrum using a microwave frequency ≈1.11GHz for two modulation amplitude values, 0.1mT and 0.3mT respectively. The result is shown below. As we can see, the use of small modulation amplitude values (MA), which limit the signal distortion, provides a low signal-to-noise ratio (S/N). On contrary, even though over-modulation affects the line shape (distortion and broadening), a slightly over-modulation can enhance the S/N ratio and then significantly enhance the sensitivity. Consequently, we chose to keep MA=0.3mT and use a smaller microwave frequency to enhance the S/N ratio. Furthermore, our work was performed on a pristine pouch cell battery. At this electrochemical state, only bulk Li-metal signal expected. However, we are agreeing with the reviewer that during charge and discharge, some sub-micrometric Li-metal particles can appear and a modulation amplitude much smaller will be necessary.*

[Figure]

*Comment#4: With a field of view of 30 mm and 512x512 pixels, a resolution of 60 µm would be*

*expected. Please specify more accurately the limiting factor of the stated resolution of 150 μm, and why the apparent resolution in the images in Fig. 3 appears to be even lower than this value.*

**Response**: *We thanks the reviewer for this relevant comment which is absolutely right. EPR images were recorded with 200x200 pixels and a field of view of 30mm which corresponds to a resolution of 150 μm.*

*The experimental details of EPR imaging is now corrected in the revised manuscript (on page 3, line 69 in red).*

*Comment#5: Section 2.1: Please provide the thickness of all the components of the cell, as shown in Fig. 2a, including current collectors. For the current collectors, it would be helpful if also the respective skin depth would be stated.*

**Response:** *Our work has been performed through a collaboration with the industrial company SAFT. SAFT does not want to provide such information on the thickness of their electrochemical components. However, in the revised manuscript, we provided some additional details about electrode materials used in this work that will allow other laboratories to reproduce our measurements.*

*Comment#6: Section 2.2 (and throughout manuscript): Please provide all units in mT.*

**Response:** *As suggested by the reviewer, we provide all units in mT throughout paper.*

*Comment#7: Section 2.3: Please specify the number of radial projections, i.e. how many projections were recorded for the images*

**Response:** *The number of radial projection used for the EPR images is 157. This value is now reported in the revised manuscript (on page 3, line 69 in red).*

*Comment#8: Line 78ff: Discussion of Dysonian lineshape is handwavy and incomplete; either provide a more accurate description or just summarize and refer to discussions in the literature*

**Response**: *We thank the reviewer for this point. This shape comes from the microwave field which excites only spins located inside the skin depth and is influenced by the non-uniformity of the microwave field in the metallic structure. For a thickness much higher than the skin depth, the EPR spectrum appears asymmetric while for a thickness lower than the skin depth, a pure Lorentzian (symmetric) shape is obtained.*

*As suggested by the reviewer, we included three references in the discussion part of the revised version of the paper (on page 4, line 87). These papers describe the Dysonian line shape of a Li-metal sample as a function of the purity and the thickness of the metallic structure.*

*We also modified the description of the Dysonian line shape in the revised version (on page 3 lines 81-85} by: "… excited.* *Consequently, the line shape is influenced by the homogeneous of the electromagnetic field in the metallic structure displaying a Dysonian line shape when a fraction of the electronic spins experiencing the microwave field and a pure Lorentzian in the opposite case for which all the electronic spins are excited* *"and deleted the sentence: "Furthermore, this phenomenon depends on the spin coherent time T2, in which electron travels on a specific distance δe, named spin depth. In metallic lithium structures, $T2 \approx 10^{-8}s$ and then when …".*

*We added "(Dysonian)" on page 4, line 86 (in red) of the revised version: "…the EPR spectrum exhibits an asymmetric (Dysonian) line shape characterized by the ratio A/B…" and "… ≫1 resulting from the*

*apparent amplitude ratio between its positive lobe and negative lobe (Gourier et al., 1989; Niemöller et al., 2018; Dutoit et al.,2021)" on page 4, lines 86-88 in red of the revised version.*

*Comment#9: Line 83: a metallic conductor does not cause a 'perturbation of the dielectric'; please state more precisely*

**Response:** *Thank you for this relevant remark. Usually, a conventional X-band EPR spectrometer is equipped with a microwave cavity similar to a metal box highly conductive. A large and over-sized metallic conductor deposited inside the microwave resonator can provide some difficulties to tune the spectrometer owing to the perturbation of the electromagnetic field inside the resonator and by the presence of the static magnetic field. Dielectric refers to microwave resonator.*

*Comment#10: Line 87: Unclear what is meant by "the dielectric is slightly perturbed". Which dielectric is perturbed?*

**Response:** *The L-band EPR spectrometer used in our work is equipped with a bridge loop-gap resonator. During our experiments, we encountered some minor issues upon inserting the pouch cell about tuning the spectrometer on. Dielectric refers to microwave resonator.*

*Coment#11: Fig. 2b: What is the origin of the line shape of the defect signal? The narrow line does not look like the first derivative of a signal one would expect from defects.*

**Response:** *The EPR spectrum obtained on the NMC811||Si-based pouch cell battery is very different from the EPR signal obtained on the NMC811||Li-metal. This first result provides evidence that L-band EPR spectroscopy can probe electrode materials wrapped up in a standard pouch cell. The EPR spectrum recorded in fig 2b exhibits two paramagnetic species: (i) a very broad line due to the NMC cathode (distortion of the baseline) and (ii) a very weak signal centered at ≈2.001. It is well known that the EPR spectrum of $SiO_2$ gives a line centered in a g-value range of 1.999-2.0018 (DOI: 10.1063/1.371996 and DOI: 10.1063/1.369464). The origin of such weak signal can be easily attributed to $SiO_2$ defects which are located inside the Si-based anode. We are agreeing that the EPR line does not exhibit the first derivative of a spectrum classically recorded in an EPR experiment. The reason is the phase sensitivity detection is not optimized because the main result expected here was only to compare the both spectroscopic signatures (NMC811||Li-metal compared to NMC811||Si-based).*

*Comment#12: Line 115: Conduction EPR images are generally not quantitative, hence quantitativity, as stated in the text, would have to be confirmed with an independent experimental technique. In this case, shielding and eddy current effects most likely affect the signal, thus the signal does not represent spin concentration at all.*

**Response***: We thank the reviewer for rising this important point.*

*We did not perform a quantitative investigation neither given the number of electron spins detected in the sample because the EPR intensity of lithium metal line does not reflect the total amount of lithium. The reasons are twofold. On one hand, only electrons of energy above the Fermi energy are unpaired (and contribute to the EPR spectrum), i.e. a fraction of about 0.4% of the electrons; on the other hand, for particles larger than several micrometers, only electrons located in the skin depth contribute to the resonance.*

*On page 6, line 126 of the first version, we used the term "spin concentration" only to describe the signal variation between areas more or less sensitive to the microwave field. What we meant here is "spin concentration seen by the microwave field". We agree with the reviewer that the "true" spin concentration cannot be obtained only by EPR imaging.*

*Comment#13: Fig. 3. The three projections cannot be congruently combined to a 3D image. What is the reason for this? Concomitant gradients? Please discuss*

**Response:** *Yes, we think the three projections can be combined to record a 3D image. However, such reconstruction requires the development of a specific algorithm and script using, for example, MATLAB.*

*Comment#14: Fig. 3. What is the reason that the noise (or apparent signal?) away from the sample, where there is definitively no Li, is larger than right around the sample?*

**Response:** *First, it is worth noting that only Li-metal from the negative electrode gives an EPR signal and contribute to the EPR image (fig.3). Secondly, conventional EPR image reconstruction method, such as filtered back projection applied may produce some artifacts in the image. This is the reason why we observe a noise signal away from the anode much larger than around the anode. However, this phenomenon does not modify the presence of higher signal produced by the metallic anode.*

*Comment#15: Fig. 3. What was the state of the battery? Was it pristine or had it been cycled?*

**Response:** *In figure 3 we provide the EPR image of the pristine battery. This image (square form) corresponds to the lithium metallic anode before cycling showing none structural modification.*

*Comment#16: Line 121 and Fit. 3: The signal from the center of the anode is not larger than at the edge of the field-of-view, where it represents exclusively noise. Therefore, there is not a "slight variation of EPR intensities", but the signal from the center of the cell cannot be conclusively distinguished from noise at all. It appears that there is only signal caused by the boundary of the cell. This should be discussed in detail.*

**Response:** *Thank you for this relevant remark. Indeed, the signal from the center part is featureless while the signal from the edge of the anode appears clearly. The reason is shown by Niemöller et al (DOI: 10.1038/s41598-018-32112-y) and comes from the edge effects. In a flat Li-metal sample, local variations of the skin depth can cause a variation of the Li-signal. The edge appear much higher than the center part caused by a shielding effect. Although Li-metal structures are also located in the middle part, their EPR signals are hidden by the signal from the edge.*

*As suggested by the reviewer, we deleted "slight" in our sentence "slight variation of EPR intensities" (on page 6, line 132 of the revised version).*

*Comment#17: Line 135: Li-metal nucleation would lead to a conduction EPR signal with substantially different line width than shown for a pristine Li metal foil. An individual localization would require special measures that further increases the acquisition time. Discuss the feasibility of such experiments in terms of the expected acquisition time, taking also into account the necessary reduction of the modulation amplitude.*

***Response:*** *Exactly, in the case of sub-micrometric metallic lithium nucleation, the line shape changes and appears much higher (apparent amplitude), symmetric (due to the skin depth) and sharp. Consequently, the time needed to record such images and locate Li-metal particles could increase significantly. This time depends on the conversion time, sweep time, number of projections, recovery delay. For example, with a conversion time of 10.24ms, 402 projections and a recovery delay of 1.5s, the time needed to record the image of a bulk Li-metal is around96min whereas with a conversion time of 20.48ms, 402 projections and a recovery delay of 1.5s, the acquisition time of micro-aggregates is around 166min.*

---

## Author Response (AR2)

**Point-by-point response to reviewers and editor for the manuscript:**

*Innovative L-band electron paramagnetic resonance investigation of solid-state pouch cell batteries,* by CE. Dutoit, R. Soleti, JM. Doux, V. Pelé, V. Boireau, C. Jordy, S. Pondaven and H. Vezin.

**Reviewer 1**

The authors thank the reviewer 1 for having accepted to published this paper in Magnetic Resonance after minor corrections.

*Comment #1: The description of the relaxation time measurements and the values measured should be included in the paper. The response to reviewer comment said that one such number was included in the revised text, but the marked-up copy shows that it was crosses out.*

*Response: Thank you for this comment. Initially and from your comment#9 (interactive discussion), we had modified the relaxation time value $T_2$ in the revised manuscript. Nevertheless, from the comment#8 of the reviewer 2, about the discussion of the Dysonian line shape, we deleted the sentence "Furthermore, this phenomenon depends on the spin coherent time T2, in which electron travels on a specific distance δe, named spin depth. In metallic lithium structures, $T2 \approx 10^{-8}s$ and then when…".*

As suggested by the reviewer 1, we included a short discussion about the relaxation time measurements in the revised version on page 4, lines 86-88.

*Comment #2: The response to reviewers said that they did not observed "local heating in the cell". It is not clear how they would have observed "local" heating "in" the cell. However, one would expect that 0.3mT modulation at 100kHz would produce overall eddy current heating of the aluminum housing and of other conductive elements. This needs to be clarified.*

*Response: Thank you for this point. The comment#2 of the reviewer 2 stated: "… Also, does this high power level cause local heating in the cell?". During measurements, we did not detect any significant heating effects. The EPR spectrum of the Li-metal anode is here similar to the one obtained for a Li-metal sample of similar size wrapped inside a Kapton pouch. Therefore, even though 0.3mT of modulation amplitude at 100kHz can produce eddy current heating the aluminum pouch cell, the effect seems to be negligible. Furthermore, taking the pouch cell with our hands, we did not note that the pouch cell was warmer than before EPR measurements.*

**Reviewer 2**

The authors have addressed most of my comments in an appropriate way. In particular, sufficient experimental details have been provided to allow reproducing the experiments. There are two points from my initial comments that I still disagree with:

The authors thank the reviewer 2 for having accepted to published this paper in Magnetic Resonance after minor corrections.

*Comment #1: Although I have misunderstood the use of the word "dielectric" (I thought the authors would refer to the sample, not the resonator as detailed in their response), I still think the word should*

*be replaced, assuming that the loop-gap resonator does not employ a dielectric? The observation that the resonator mode changes is undisputed, my comment refers only to the use of the word "dielectric".*

**Response:** *Thank you for this comment.* As suggested by the reviewer 2, we replaced the word "dielectric" by "resonator" and we modified the sentence: "Indeed placing a large metallic conductor inside a standard X-band microwave cavity can cause serious perturbations of the dielectric making it impossible to tune the cavity "(on page 4, lines 94) by:

Indeed, placing a large metallic conductor inside a standard X-band microwave cavity can cause serious perturbations of the resonator making it impossible to tune the cavity.

Also, we modified the sentence: "As a result, the dielectric is slightly perturbed by inserting the metallic sample (see Figure 1(b))" (on pages 4-5, lines 98-99) by:

As a result, the microwave resonator is undisturbed by inserting the metallic sample (see Figure 1(b))

*Comment #2: I still fail to imagine a 3D geometry that would explain all three 2D images at the same time. Considering for simplicity, but without loss of generality, that the image can be represented by four bars along the edges, as suggested by the xz image. Then the yz image suggests that all four bars must be located within a plane that is slightly tilted. Such a tilted geometry would explain why the xy image is not simply a single line as well. However, the actual xy image, consisting mainly of two bars that are not parallel to each other, would not result in the displayed narrow line in the yz image. This discrepancy should be discussed in the text.*

**Response:** *Thank you for this point. A 3D image reconstruction from three 2D images (not recorded during measurements, but reconstructed from 2D images) can provides a detailed location of metallic lithium aggregates and lithiated graphite complexes but without providing more chemical information. It can be seen as a complementary image.*

*All three 2D images were recorded without modifying the physical position of the pouch cell inside the resonator. Therefore, the "irregularities" observed on xz, xy and zy images come from the metallic anode and/or the current collectors. From all three 2D images, it can be seen the pouch cell is tilted in the yz plane and the xy plane. However, the "line" observed in the yz plane, corresponding to the anode electrode, indicates that the pouch cell is aligned in the x direction.*

*We are in agreement with the reviewer that our xy displays a weak irregularity around X=-6mm and Y=-3mm. The origin of this irregularity comes from the current collector with a semi-circle shape clearly visible in the xz image (X=-6mm and Z between -6mm and -3mm). In the limit of resolution, this current collector is aligned to the yz plane but not in the xz plane (as indicated by the dashed square in the figure below). This point is clarified in the revised version on page 7.*

The semi-circle shape is also clearly visible at X=-6 mm and Y=-3 mm in the XY plane but not in the YZ plane indicating that the pouch cell is aligned in the X direction

*Furthermore, the metallic anode is not a perfect square but exhibits some surface roughness which explain why we do not observe perfect parallel bars.*

NMC‖Li⁰ L-band EPR imaging Pristine